# Mathematical Modeling of The Challenge to Detect Pancreatic Adenocarcinoma Early with Biomarkers

## Alex Root

Molecular Biology Program, Memorial Sloan Kettering Cancer Center, New York, NY 10065, USA; rveta@mskcc.org

**Abstract:** Pancreatic ductal adenocarcinoma (PDAC) is an aggressive tumor type and is usually detected at late stage. Here, mathematical modeling is used to assess the feasibility of two-step early detection with biomarkers, followed by confirmatory imaging. A one-compartment model of biomarker concentration in blood was parameterized and analyzed. Tumor growth models were generated from two competing genomic evolution models: gradual tumor evolution and punctuated equilibrium. When a biomarker is produced by the tumor at moderate-to-high secretion rates, both evolutionary models indicate that early detection with a blood-based biomarker is feasible and can occur approximately one and a half years before the limit of detection by imaging. Early detection with a blood-based biomarker is at the borderline of clinical utility when biomarker secretion rates by the tumor are an order of magnitude lower and the fraction of biomarker entering the blood is also lower by an order of magntidue. Regardless of whether tumor evolutionary dynamics follow the gradual model or punctuated equilibrium model, the uncertainty in production and clearance rates of molecular biomarkers is a major knowledge gap, and despite significant measurement challenges, should be a priority for the field. The findings of this study provide caution regarding the feasibility of early detection of pancreatic cancer with blood-based biomarkers and challenge the community to measure biomarker production and clearance rates.

**Keywords:** pancreatic ductal adenocarcinoma; early detection; biomarker; one compartment model; mass spectrometer; ELISA; shedding rate; clearance rate; limit of quantitation

## 1. Introduction

Pancreatic ductal adenocarcinoma (PDAC) is detected at late stage and has a dismal five-year survival [1]. Early detection can provide a window of opportunity and dramatically increase cure rates [2]. Blood-based biomarkers are a potential method for early detection [3]. However, development of biomarkers has proven to be enormously challenging [4]. PDAC was long believed to follow a gradual evolution model with a particular sequence of mutations (KRAS, followed by CDKN2A, then P53 and SMAD4) with landmark work by Yachida and colleagues concluding that at least a decade is required between the occurrence of the first mutated clone, and an additional 5 years for the acquisition of metastatic capability, after which patients usually die within 2 years [5]. Recent work analyzed patterns of chromosomal alterations to challenge this view, proposing that tumor evolution may not be sequential and gradual, and that, instead, it may occur rapidly, following a punctuated equilibrium model of evolution [6]. Mathematical modeling of cancer has provided numerous insights into tumor progression and therapy [7]. Therefore, I sought to use mathematical modeling to study early detection of PDAC, by considering both the gradual evolution and punctuated equilibrium models.

For mathematical modeling of early detection, Swanson and colleagues developed a one-compartment model to relate prostate cancer tumor size with PSA volume, assuming that both cancer and normal cells secrete PSA [8]. Lutz and colleagues used a similar one-compartment model

and provided estimates of minimally detectable tumor sizes and earliest detection times based on blood tumor biomarker assays using physiological data for prostate cancer (PSA) and ovarian cancer (CA125) [9]. Hori and colleagues refined this model for ovarian cancer (CA125), finding that a tumor could grow unnoticed for approximately 10 years and reach a size of 25.36 mm diameter (8.5 cm$^3$) before becoming detectable by clinical blood assays [10]. In a follow-up study, Hori and colleagues created a mouse model with luminescent orthotopic A2780 cells that shed secreted alkaline phosphatase (SEAP) to monitor how blood levels of SEAP correspond to tumor size in a two-compartment model [11]. Furthermore, they also used allometric scaling equations and data from prostate cancer (PSA) to find that detection and discrimination of aggressive vs nonaggressive tumors could be determined with blood-based tests alone as early as 7.2 months and 8.9 years, respectively [11].

Application of mathematical modeling to early detection of PDAC might provide additional insights and guide future experiments to meet the challenge of an early detection test for PDAC that has eluded investigators for decades [12]. Several factors serve to motivate a theoretical exploration of early detection for PDAC using blood-based biomarkers, including: the historical difficulty in finding biomarkers for early detection of pancreatic cancer [2]; recently available *in vivo* data from PDAC patients [13]; and conflicting theories for the timing of PDAC genomic evolution [5,6]. In this study, I use a previously developed, one-compartment model of biomarker secretion to provide PDAC-specific estimates of earliest detection times and prioritize parameters for experimental measurements, assuming either a gradual model of decade-long PDAC progression [5] or an alterative model of PDAC progression (punctuated equilibrium) [6]. I also investigate the sensitivity of earliest detection times by varying model parameters over orders of magnitude. Finally, I highlight the areas with critical knowledge gaps to guide future experiments.

## 2. Results

### 2.1. Development of a One-Compartment Biomarker Model for PDAC

To study whether early detection of PDAC is feasible with molecular biomarkers, I parameterized and analyzed a one-compartment model motivated by previous work in prostate and ovarian cancers [8,9]. The components of the one-compartment model are shown in Figure 1. The model requires an equation for tumor volume over time and I curated the PDAC literature to develop such an equation, with parameters and values given in Table 1. In the most comprehensive study examined, Haeno and colleagues analyzed human PDAC tumors measured at diagnosis and autopsy [13]. They concluded that an exponential growth model was appropriate and unable to fit logistic growth equations, which is surprising and highlights the rapid lethality of PDAC [13]. At diagnosis, they found the average tumor to be approximately 26.5 cm$^3$. To estimate the time frame over which this growth occurred, studies on the genomic evolution of PDAC were examined [5,6]. These studies present contrasting models of genomic evolution, a gradual evolution model or punctuated equilibrium model. In the former, there is a T1 stage during which tumors begin to transition from infiltrating to metastatic, and a T2 stage at which they begin metastatic colonization [5]. The T2 stage was estimated to occur at the approximate time of tumor diagnosis when the average tumor volume is 26.5 cm$^3$. From these estimates, the exponential growth constant is 0.003 day$^{-1}$, which corresponds to a tumor volume doubling time of 209 days. This is in reasonable agreement with a previously published estimate of 159 days (64–255 day range) [14]. For the punctuated equilibrium model, phase 1 growth was assumed to happen over the tumor growth stages of T1: PanIN-Infiltrating period at 11.7 years when the tumor reaches a 10 mm radius (0.5 cm$^3$), which is an estimated size that causes nutrient limitations that require tumor progression for further growth. During phase 2, growth is assumed to rapidly increase due to simultaneously genomic events; i.e., punctuated equilibrium, where it then catches up to the gradual evolution model by T2: Infiltrating-Metastatic period. During phase 3 of tumor growth, the punctuated equilibrium model is assumed to slow down and grow at the same exponential rate as the gradual evolution model. Baseline growth rates are shown for both models in Figure 2.

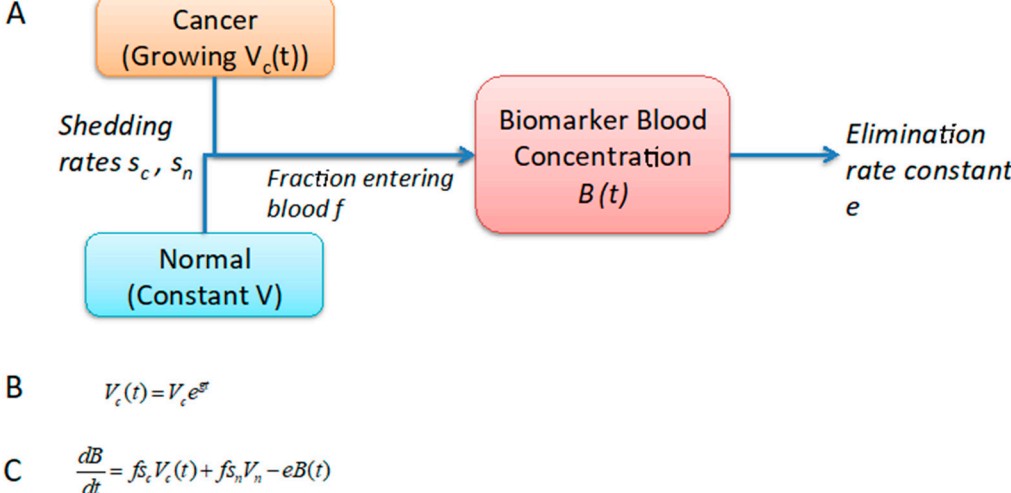

A

**Cancer**
**(Growing $V_c(t)$)**

*Shedding rates $s_c$, $s_n$*

*Fraction entering blood $f$*

**Biomarker Blood Concentration $B(t)$**

*Elimination rate constant $e$*

**Normal (Constant $V$)**

B    $V_c(t) = V_c e^{gt}$

C    $\dfrac{dB}{dt} = fs_c V_c(t) + fs_n V_n - eB(t)$

**Figure 1.** Schematic of the one-compartment mathematical model. (**A**) The one-compartment model models biomarker concentration in blood, where the biomarker can be shed by both normal and cancer cells and is eliminated at a constant rate. (**B**) The tumor growth equation is simple exponential growth based on autopsy measurements of tumor volumes, including primary and metastases. (**C**) Ordinary differential equation for biomarker production consisting of influx from tumor shedding and normal cell shedding, minus clearance from blood.

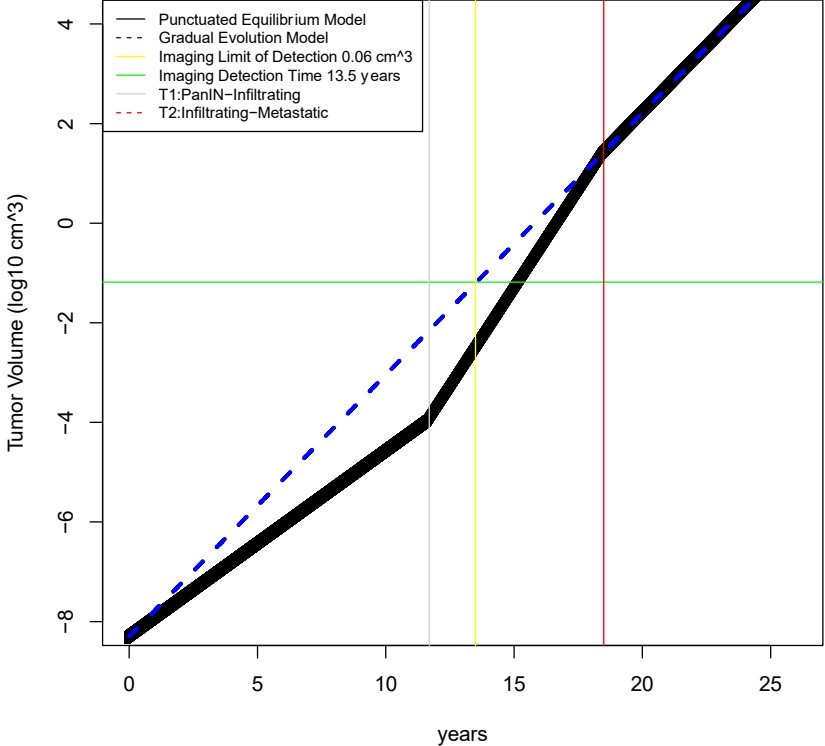

**Figure 2.** Timecourse of tumor growth in gradual evolution vs punctuated equilibrium models. In the gradual evolution model, tumor growth is assumed to be exponential from the single founder clone all through the disease course. In the punctuated equilibrium model, the tumor is assumed to exhibit exponential growth in 3 phases, with the middle phase representing catastrophic genomic events that rapidly accelerate growth. Medical imaging is assumed to have a lower limit of detection of 5 mm$^3$ (2.125 mm diameter) which corresponds to 13.5 years of tumor growth in the gradual evolution model.

**Table 1.** Tumor growth model parameters, input value ranges (if applicable), and sources.

| Parameter | Symbol | Unit | Baseline | Range | Source |
|---|---|---|---|---|---|
| Diameter of primary tumor at diagnosis (cohort 1: 69% Stage I–III, 31% StageIV) | 2r | cm | 3.7 | 2.8–4.2 | Haeno [13] |
| Diameter of primary tumor at diagnosis (cohort 2: 100% Stage I–III) | 2r | cm | 3 | 2.5–4 | Haeno [13] |
| Volume of primary tumor at diagnosis (cohort 1: 69% Stage I–III, 31% StageIV) | V | $cm^3$ | 26.5 | 11.5–38.8 | (calculated from Haeno [13]) |
| Volume of primary tumor at diagnosis (cohort 2: 100% Stage I–III) | V | $cm^3$ | 14.1 | 8.2–33.5 | (calclulated from Haeno) |
| Tumor volume doubling time | TVDT | days | 159 | 64–255 | Furukawa [14] |
| Density of cancer cells in solid tumor tissue | $d_c$ | $cells/cm^3$ | $2 \times 10^8$ | (na) | Lutz [9] |
| Density of pancreatic cancer cells in solid tumor tissue | $d_c$ | $cells/um^3$ | $2.85 \times 10^3$ | (na) | Kisfalvi [15] |
| Volume of a single cancer cell | $v_c$ | $mm^3$ | $5 \times 10^{-6}$ | (na) | Lutz [9] |
| Time to reach infiltrating capability—gradual model * | $T_{3gm}$ | year | 18.5 * | 12–25 | Yachida [5] |
| Growth rate of gradual model | g | $day^{-1}$ | 0.003316444 | (na) | Calculated |
| Tumor volume doubling time for gradual model | $TVDT_{gm}$ | days | 209 | (na) | Calculated |
| Time to reach infiltrating capability—punctuated equilibrium model Phase 1 | $T_{3pem}$ | year | 11.7 | 10–23 | (estimated from Notta [6] and Yachida [5]) |
| Tumor diameter at infiltrating capacity punctuated equilibrium model | 2r | cm | 0.01 | (na) | assumption |
| Growth rate of punctuated equilibrium model, phase 1 | $g^1$ | $day^{-1}$ | 0.001089422 | (na) | Calculated |
| Growth rate of punctuated equilibrium model, phase 2 | $g^2$ | $day^{-1}$ | 0.007147335 | (na) | Calculated |
| Growth rate of punctuated equilibrium model, phase 3 | $g^3$ | $day^{-1}$ | 0.003316444 | (na) | Calculated |
| Tumor volume doubling time for punctuated equilibrium model during phase 2 | $TVDT_{pem}$ | days | 97 | (na) | Calculated |
| Average primary tumor volume at autopsy | V | $cm^3$ | 524 | (na) | (calculated from Haeno [13]) |
| Average number and size of metastatic tumors at autopsy | V | $n, cm^3$ | 100, 4.19 | (na) | (calculated from Haeno [13]) |
| Average sum of primary and metastatic tumor volumes at autopsy | V | $cm^3$ | 943 | (na) | (calculated from Haeno [13]) |

* Calculated from time of initiated tumor cell to parental clone (T1) plus time from parental clone to subclones with metastatic ability (T2) corresponding to PanIN3.

Following development of the tumor growth models, the remaining parameters for the one-compartment model were parameterized with literature curation and estimation. In these steps, much greater uncertainty was encountered and model parameters had to be borrowed from prostate or ovarian cancer models; Table 2. It was assumed that PDAC biomarkers would be shed by both cancer and normal cells. It was also assumed that the biomarker would be PSA-like and therefore, for healthy individuals a concentration of 1.38 ng/mL was used with a disease cutoff of 4 ng/mL. The parameters with the greatest uncertainty include the following: shedding rate constant of biomarkers in cancer and normal tissue, and the fraction of shed biomarker that reaches the blood from the interstitium.

These parameters were investigated over several orders of magnitude. To be considered potentially useful, a biomarker was required to reach the cutoff with its earliest detection time preceding the T2: infiltrating-metastatic transition at 18.5 years, at which time most PDAC tumors are assumed to be detected. An ideal biomarker would have an earliest detection time closer to T1:PanIN-cancer transition at 11.7 years, which is about a year and a half earlier than the time that a tumor is estimated to reach the lower limit of detection by medical imaging.

**Table 2.** Biomarker concentration model parameters, input value ranges, and sources.

| Parameter | Symbol | Unit | Baseline | Range | Source |
|---|---|---|---|---|---|
| Biomarker concentration in blood over time | B(t) | ng/mL | TBC * | TBC * | (na) |
| Volume of blood in a typical male or female person | $V_{M,F}$ | mL | 3150, 3825 | (na) | Lutz [9] |
| Shedding rate of biomarker from cancer cells | $s_c$ | $ng(10^5 cells)^{-1}(day)^{-1}(mL)^{-1}$ | 2.1 | 2.1–200 | Lutz [9] |
| Shedding rate of biomarkers from tumor | $s_c$ | $ng/cm^3 (day)^{-1}(mL)^{-1}$ | 4200 | 420–42,000 | calculated from Lutz |
| Fraction of biomarker that enters blood from interstitium | f | (na) | 10% | (na) | Lutz [9] |
| Steady State biomarker concentration in healthy controls | $B_n$ | ng/ml | 1.38 | (na) | Lutz [9] |
| Influx of biomarker shed from normal cells | $V_n S_n$ | $ng(day)^{-1}(mL)^{-1}$ | 17.75 | (na) | calculated |
| Biomarker elimination rate from blood | e | $day^{-1}$ | 1.286 | 0.129–12.86 | Swanson [8] |
| Limit of detection for molecular assay | $LOD_{assay}$ | ng/mL | 0.1 | 0.01–1 | Zhang [16] |
| Limit of detection for imaging | $LOD_{imaging}$ | $mm^3$ | $5 mm^3$ | (na) | Hori [11] |
| Time of detection for imaging assuming gradual evolution model | $T_d$ | years | 11.4 | (na) | calculated |

* TBC is to be calculated from the model using the other parameters.

*2.2. Determination of Earliest Detection Times for Baseline Parameter Values*

Using the best estimates for baseline model parameter values, early detection using a molecular biomarker was determined to be feasible and clinically useful. For the gradual evolution model, earliest detection time occurs at 11.81 years when the tumor has reached only 0.01 $cm^3$; Figure 3. For the punctuated evolution model, earliest detection time occurs at approximately the same time. Detection at this time represents a disease stage approximately 5 years before metastasis occurs. Detection at this time is also below the limit of detection for medical imaging, and therefore, patients with a postive biomarker test would likely need to undergo annual imaging to confirm disease. Overall, calculations with these baseline values are favorable results for early detection with a molecular biomarker. To determine how sensitive these results are to variations over biologically plausible parameter values, one-way sensitivity analyses were performed.

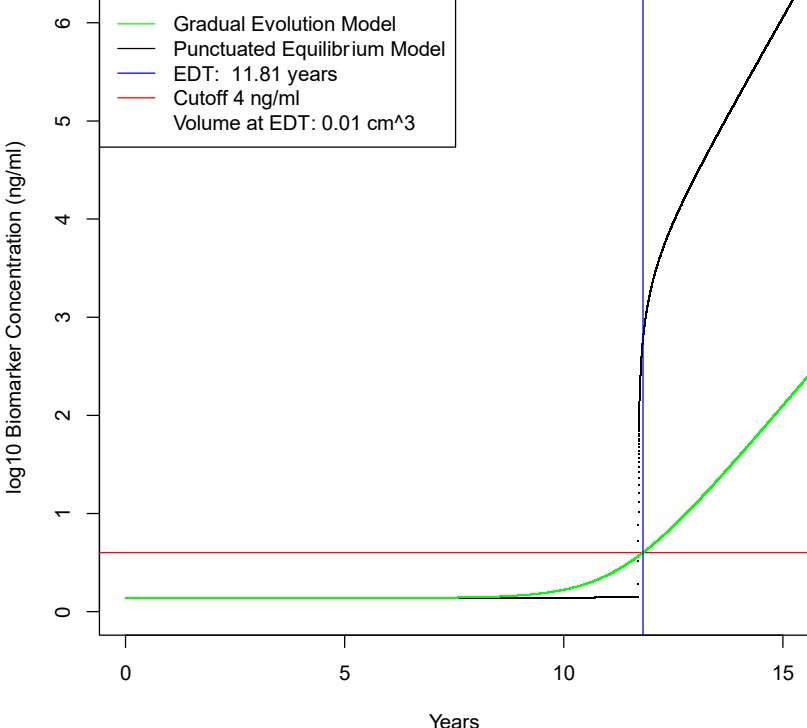

**Figure 3.** Earliest detection times in gradual evolution vs punctuated equilibrium models. In the first decade of tumor growth, the biomarker concentration in blood is due to secretion by normal cells. After this, biomarker concentration is increasingly due to tumor production. At approximately 11.8 years, blood biomarker levels reach 4 ng/mL, which is the assumed cut-off point for early detection with a blood test. This is approximately 1.5 year earlier than the limit of detection for medical imaging of 5 mm$^3$.

*2.3. Sensitivity Analysis of Model Parameters on Earliest Detection Times*

A total of five parameters were investigated using one-way sensitivity analysis: volume of primary tumor at diagnosis, time T2 of invasive-to-metastatic transition, shedding rate of biomarkers from the tumor, fraction of biomarker that enters the blood from the interstitium, and biomarker elimination rate constant across values given in Table 3. The first two parameters, volume of primary tumor at diagnosis and time at invasive-to-metastic transition, are given in the PDAC literature with relatively precise values. However, the remaining three parameters have uncertain values in PDAC and were varied over orders of magnitude. Variation in tumor volume at diagnosis has very little effect on earliest diagnosis time, reflecting the precision with which this parameter is specified; Figure 4A. The time for invasive-to-metastic transition has a larger effect on earliest detection time, but even for the worst case simulated, earliest detection is still feasible and likely to be clinically useful; Figure 4B. Variations in the tumor shedding rate constant have a large effect on earliest detection times; Figure 4C. The fraction of secreted biomarker entering blood needs to be at least 4% otherwise a blood-based biomarker does not outperform imaging; Figure 4D. Finally, if the elimination rate constant increases from 1.286 at baseline to above 5 then a blood-based biomarker does not outperform imaging; Figure 4E.

**Table 3.** Sensitivity analysis: model parameter means, standard deviations, and probability distributions.

| Parameter | Symbol | Baseline Value | Minimum | Maximum |
|---|---|---|---|---|
| Volume of primary tumor at diagnosis | $V_c$ | 3cm diameter 2.8–4.2 Haeno [13] | 2.8 diameter = 11.49 cm$^3$ | 4.2 diameter = 38.79 cm$^3$ |
| T2: time at invasive-to-metastatic transition | $t_3$ | 18.5 +/− 3.4 years Yachida [5] | 15.1 | 21.9 |
| Shedding rate of biomarkers from tumor | $s_c$ | 4200 ng/cm$^3$ $(day)^{-1}(mL)^{-1}$ over a 100× range Lutz [9] | 420 | 42,000 |
| Fraction of biomarker that enters blood from interstitium | $f$ | 0.10 Lutz [9] | 0.001 | 0.20 |
| Biomarker elimination rate constant from blood | $e$ | 1.286 day$^{-1}$ over a 100× range Swanson [8] | 0.1286 | 12.86 |

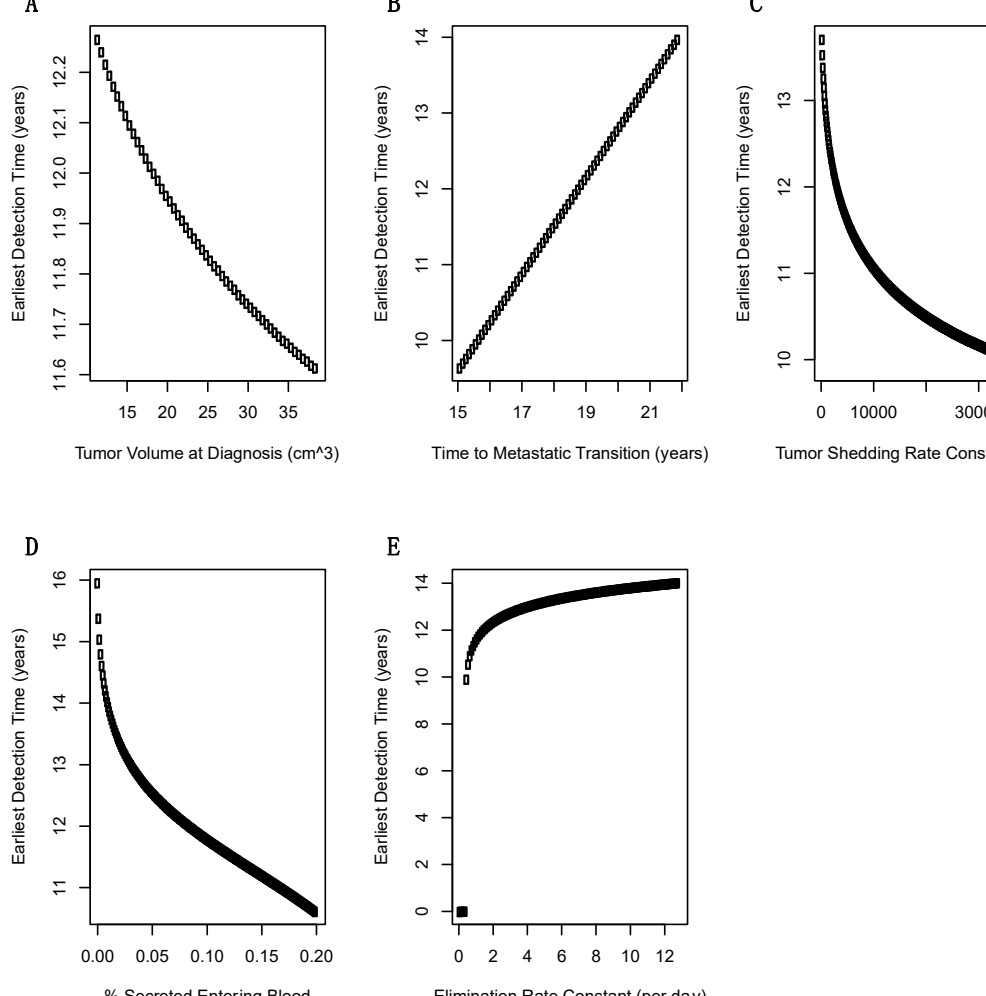

**Figure 4.** Sensitivity of minimum detection times over one-dimensional scans of model parameters. (**A**) For the range of tumor volumes at diagnosis there is minimal effect on earliest detection time. (**B**) There is a linear dependence between time to metastatic transition (T2) and earliest detection time that has an important effect on when biomarker screening should begin. (**C**) Tumor biomarker shedding rate constant has a exponential effect on earliest detection time and is therefore, a critical parameter. (**D**) The fraction of secreted biomarker entering blood also has an exponential effect on earliest detection times. (**E**) The blood elimination rate constant has important effects that are generally mild at the most likely values of >2 day$^{-1}$.

### 2.4. Calculation of Earliest Detection Times for Two Unfavorable Scenarios

The one-way sensitivity analysis did not reveal many areas of parameter space where early detection would not be clinically useful, so two different scenarios were investigated assuming increasingly unfavorable combinations of multiple parameters that are plausible given the knowledge gaps in biomarker secretion rates for PDAC. In scenario 1, tumor secretion is 1/2 lower and the fraction of biomarker entering the blood is 1/2 lower than baseline values; Table 4. This results in an earliest detection time of 13.1 years; Figure 5. This represents clinically useful early detection because the tumor is at early stage and the detection occurs earlier than by imaging. In scenario 2, tumor secretion is reduced 10x and the fraction entering the blood is also reduced by 10x; Table 4. Here, the earliest detection time deteriorates by almost 2.5 years and the tumor is on the border of the metastatic stage; Figure 5. This is earlier than current diagnoses and is probably clinically useful, although less so than imaging. These scenarios demonstrate that order of magnitude changes in model parameters may affect whether blood-based biomarkers are clinically useful. Therefore, without accurate estimates of biomarker secretion rates by the tumor and penetration rates into blood, the feasibility of early detection cannot be confidently asserted.

**Table 4.** Two-way Parameter Variation Scenarios.

| Early Detection Scenario | Tumor Growth Rate Constant ($g$) day$^{-1}$ | Tumor Secretion Rate Constant ($S_c$) ng/cm$^3$ (day)$^{-1}$(mL)$^{-1}$ | Normal Secretion Influx ($V_n S_n$) ng/day | Biomarker Elimination Rate Constant ($e$) day$^{-1}$ | Fraction of Biomarker Entering Blood ($f$) |
|---|---|---|---|---|---|
| Baseline | 0.003315953 | 4200 | 17.75 | 1.286 | 10% |
| Scenario 1 | 0.003315953 | 1/2× lower 2100 | unchanged 17.75 | unchanged 1.286 | 1/2× lower 5% |
| Scenario 2 | 0.003325953 | 10× lower 420 | unchanged 17.75 | unchanged 1.286 | 10× lower 1% |

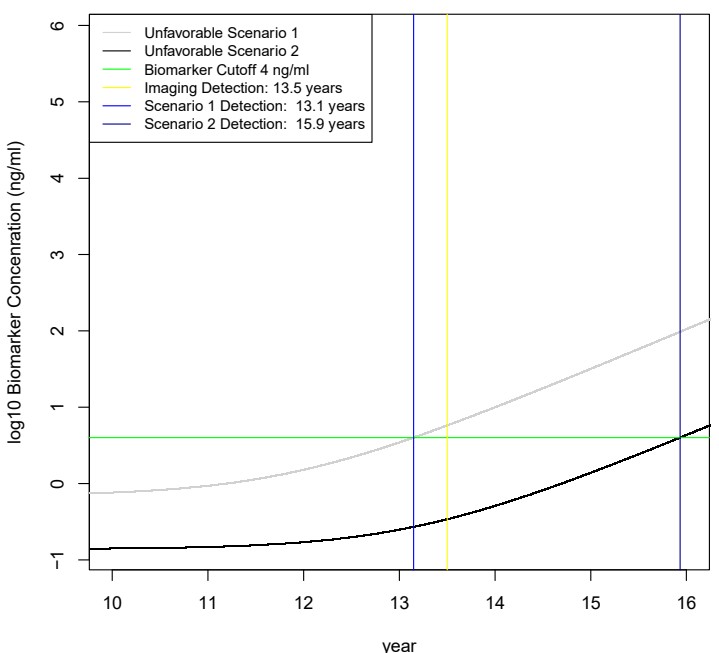

**Figure 5.** Investigation of two unfavorable scenarios on earliest detection times. Scenario 1 is 1/2 reduction in tumor biomarker secretion rate and 1/2 reduction in fraction of biomarker entering the blood, which results in an earliest detection time of 13.1 years that is slightly earlier than the 13.5 year threshold for detecting disease by imaging. Scenario 2 is a 10-fold reduction in the tumor biomarker secretion rate and a 10-fold reduction in the fraction of biomarkers entering the blood, which results in an earliest detection time of 15.9 years. Therefore, order of magnitude decreases in tumor secretion rates and fraction entering the blood result in significantly later detection times than imaging.

## 3. Discussion

After performing literature curation and specifying a one-compartment model for early detection with a blood-based molecular biomarker, calculations with baseline values showed that early detection is feasible and clinically useful. Earliest detection times under baseline scenarios allow for detection 5 years before the start of tumor metastasis. However, most model parameters were taken from PSA measurements in prostate cancer, which typically is less aggressive than PDAC, so it is uncertain whether these results are probable, plausible, or inaccurate. Sensitivity analysis was performed across orders of magnitude for most model parameters. Parameters such as the biomarker clearance rate from blood and time to T2: infiltrating-metastatic transition did not have significant effects on the feasibility or clinical utility of early detection. The results generally support the feasibility of early detection with a molecular biomarker that is similar to PSA; however, there are significant regions of parameter space where early detection with a blood-based biomarker occurs later than by imaging. For example, if the fraction of biomarker entering the blood from the interstitium is 1% and biomarker secretion by tumor cells is 10x below baseline, then imaging would detect a tumor more than two years earlier than a blood-based biomarker. This is probably still clinically useful because a blood-based biomarker test may have significantly lower economic costs than imaging and still detect PDAC tumors at earlier stages, which are known to have higher survival [17].

In this study, tumor growth was modeled by exponential equations. This was motivated by Haeno and colleagues, who analyzed human PDAC tumors measured at diagnosis and at autopsy in the largest and most comprehensive study to my knowledge [13]. They concluded that an exponential growth model was appropriate, linear models had poor fits, and were unable to fit logistic equations [13]. This suggests that PDAC is a very aggressive cancer that may not be constrained by nutrient availability that is observed in other tumor types [18]. A widely used model for tumor growth that is constrained by a type of logistic curve is known as the Gompertz equation [19]. Fitting this model requires at least three data points because it contains an additional parameter (growth deceleration). Modeling representations that include aspects of tumorigenesis such as angiogenesis, contact inhibition, nutrients, and hypoxia are much richer formulations that would result in more mechanistic insights [20]. Similarly, agent-based and multiscale models incorporate behavior at the cell, tissue, and organ levels that provide for greater insights [21]. The approach used here is comparatively simple and based on limited data from PDAC patients. The one-compartment biomarker model is clearly unrealistic at later stages because biomarker levels grow exponentially in the model, which they do not do in patients. There is much room for further research into modeling PDAC growth and biomarker concentrations, for example by using time-course imaging of mouse models, coupled with advanced sensors for hypoxia, and interpolation to human growth models using allometric scaling [11]. It would also be advantageous to combine stochastic evolutionary dynamics models with growth models in order to determine the timing and identity of mutated nucleic acids that may be shed into blood.

The one-compartment model used here is applicable to other tumor times. As published parameter values only exist for prostate and ovarian cancers, it is unclear what the extent of variation is in the model parameters (shedding rates, fraction entering blood, elimination). It may be that leukemias and lymphomas have the highest fraction of biomarkers entering blood because of their direct connections to circulation, and, conversely, that tumors of the brain would have the lowest fraction entering blood because of the blood-brain barrier. A tumor may directly secrete a biomarker into blood or indirectly cause biomarkers secreted by normal cells to leak into the blood. This effect is due to disruption of the ECM by tumor invasion, or due to angiogenesis. Leakage of biomarkers may be due to wounding or inflammation, and therefore, be less specific for cancer. Benign pancreatic diseases such as pancreatitis may leak biomarkers and therefore, tumor vs benign disease controls must be included in biomarker studies. Alternatively, leakage may be due only to tumor secretion and therefore, highly cancer specific, resulting in the release of a protein with good biomarker properties, such as high concentrations. Biomarker secretion by normal cells may be age-dependent and therefore, greater

accuracy can be achieved by setting age-dependent cut-offs in ELISA assays, or modeling population structure explicitly.

There are aspects of the PDAC microenvironment that may make the parameters used in this investigation inaccurate because they were borrowed from prostate and ovarian cancer models, which have different microenvironments. For example, primary PDAC tumors are known to be highly desmoplastic with high interstitial fluid pressure and vasculature collapse, which results in hypoperfusion [22]. This hypoperfusion results in poor drug delivery and treatment failure [22]. Hypoperfusion due to high interstititial pressures might result in a significant difference in the fraction of the biomarker entering the blood (model parameter *f*). What is the net result of this phenomena? A higher interstitial pressure might drive more biomarker into the blood, making *f* higher than in prostate or ovarian. On the other hand, because the tumor is poorly perfused, there is an overall lower surface area of blood vessels for biomarker entry, which would make the entry fraction lower. Because earliest detection time estimates depend heavily on this parameter, this phenomena requires additional investigation in PDAC tumors. In this investigation, the fraction of biomarker entering plasma was modelled at only 10% levels, for which early detection is feasible. If this parameter is actually an order of magnitude or more lower because of hypoperfusion or another factor, then early detection with a blood-based biomarker may be borderline if other parameters are also significantly less favorable.

Tumor growth has been modelled with sophisticated partial differential equation models that provide enhanced realism by modeling oxygen and nutrient-limiting gradients affected by abundant stroma [23]. These effects can be incorporated in future models. There are additional PDAC-specific effects that may need to be modelled. For example, PDAC cells are known to scavenge for amino acids to fuel growth in a process called macro-pinocytosis [24]. Therefore, it may be that tumor biomarker secretion is not constant and might decrease due to macro-pinocytosis. This may provide a reason why biomarker concentrations do not increase monotonically with disease burden. Similarly, biomarker clearance from blood may not follow linear kinetics due to a variety of effects. Generally, clearance of proteins from blood is a poorly understood process, even for major blood proteins such as albumin [25]. These phenomena deserve prioritization for measurement in PDAC mouse models.

Analyses presented here used tumor-specific parameters, which are relatively crude given the considerable heterogeneity among cancer types. A logical next step is to develop models for cancer subtypes. Over the long run, it is essential to make personalized models of biomarkers in order to optimize screening strategies. For example, a future research goal is to associate genetic sequences with parameters in a biomarker model, such as variations in extracellular matrix genes that might alter the fraction of biomarker entering blood, or variations in biomarker gene promoter sequences might affect secretion rates by a tumor. These personalized models may then inform monitoring rates, or, indeed, even which different biomarkers are monitored, depending on personalized characteristics.

## 4. Methods

### 4.1. Estimation of Tumor Growth Rates

Literature curation was performed to determine an appropriate tumor growth model (linear, exponential, power law, logistic, Gompertz, or other) and its parameters [19]. An exponential model was selected, and its growth model equation is:

$$V(t) = V(0)e^{gt} \tag{1}$$

where $V(t)$ is the tumor population over time, $V(0)$ is the initial tumor population, and $g$ is the growth rate. Literature curation was performed manually in order to determine estimates of these parameters; Table 1. Tumor volume was modelled rather than tumor population because the limit of detection is known for imaging. Tumor population sizes were converted to volumes assuming a constant volume per cell, spherical tumor, and no dependence of cell volume upon population; i.e., compression effects were ignored. The volume of a single PDAC cancer cell used is $5 \times 10^{-6}$ mm$^3$ reported by Lutz and

colleagues [9]. An initial tumor volume $V(0)$ was calculated assuming an initial tumor population of a single cell. To calculate the tumor growth rate, a simple formula was used, from Mehrara and colleagues [26]: $g = ln(V2/V1)/(T2 - T1)$. For the gradual evolution model presented by Yachida and colleagues [5], the average tumor grows for 18.5 years when it reaches the transition from invasive to metastatic growth and metastasis has occured. I estimate that this is approximately the time when most tumors are detected at advanced stage [1]. In a recent dataset, the average tumor volume at diagnosis is 26.5 cm$^3$ and 39 out of 103 patients had detectable metastatic tumors [13]. Therefore, I estimate that a reasonable time period for tumor growth until diagnosis averages 18.5 years with corresponding tumor sizes averaging 26.5 cm$^3$.

Using these estimates results in an exponential growth rate constant of 0.003 day$^{-1}$. For the punctuated equilibrium model, Notta and colleagues gave only a qualitative description [6] that I parameterized into three growth phases as follows: In phase 1, tumor growth occurs very slowly, consisting of genetic drift until a tumor volume of 1 mm$^3$, at which time angiogenesis is necessary for tumor growth and infiltration [27]. This is assumed to take 11.7 years, which is the identical amount of time as the gradual evolution model because the early genetic alterations are similar between both models, resulting in a calculated growth rate of 0.002 day$^{-1}$. However, in phase 2, the growth rate dramatically increases in the punctuated equilibrium model due to catastrophic chromosomal events that confer a large fitness benefit [6]. For simplicity, I assume that this occurs until the T2 time given by Yachida when metastasis is thought to begin. In phase 3, metastasis occurs and the growth rate in the punctuated equilibrium model is assumed to be identical to gradual growth model; i.e., tumor volume doubling time is on average 209 days.

## 4.2. Estimation of Biomarker Concentration in Blood

To estimate biomarker concentration in blood, a one-compartment model consisting of biomarker production by normal cells, biomarker production by cancer cells, and biomarker elimination from blood was assumed to be well-mixed and homogenous, motivated by work from Swanson and colleagues [8] and Lutz and colleagues [9], who determined parameter values from studies of the biomarkers PSA in prostate cancer or CA125 in ovarian cancer. An ordinary differential equation representing the one-compartment model was used:

$$\frac{dB}{dt} = fs_c V_c(t) + fs_n V_n - eB(t) \tag{2}$$

where $B(t)$ is the concentration of the biomarker in blood, $f$ is a coefficient for fraction of biomarker reaching blood from interstitium, $s_c$ and $s_n$ are the secretion rates from cancer and normal cells, $V_c$ and $V_n$ are cancer and normal cells volumes, $e$ is the elimination rate from blood. The volume of normal cells is assumed to be constant. Biomarker secretion and elimination rates are also assumed to be constant. Tumor growth is assumed to be exponential with values; Table 1. Initial tumor volume is assumed to consist of a single cell with a volume of $5 \times 10^{-9}$ cm$^3$. Values for the one-compartment model parameters with their units and sources in the literature; Table 2. To solve the ordinary differential equation and find the timecourse of biomarker concentration in blood, numerical integration was implemented using the deSolve package in the R programming language [28].

## 4.3. Sensitivity of Earliest Detection Times over Variations in Model Parameters

The purpose of sensitivity analysis is to determine how variations in model parameters affect quantities of interest, such as model output, convergence, stability, and other quantities [29]. Sensitivity analysis was peformed using one-dimensional parameter scans [30]. For each parameter, baseline value and ranges are specified in Table 3. Parameter scan ranges were chosen for distributions to achieve parameter spaces over two orders of magnitude except where refined estimates of variance exist, which is generally similar to the 100× ranges explored by Lutz and colleagues [9]. Sensitivity analysis was performed only for the gradual evolution model.

*4.4. Investigation of Three Scenarios on Earliest Detection Times*

To investigate how earliest detection times vary across biologically plausible values for model parameters, three different parameter combinations were investigated corresponding to baseline values, unfavorable, and favorable biomarker secretion values for early detection. Due to secretion of biomarkers by normal cells, the lower limit of quantitation of an assay, such as an ELISA or PRM, may be exceeded in normal patients. Therefore, the lower limit of quantitation is not discriminatory for cancer vs. normal. In these cases, determination of cancer vs normal is assumed to follow a cutoff level for PSA (4 ng/mL) because biomarker and normal secretion rates were obtained for PSA [9]. There is an age-dependence on normal PSA concentration [31]. However, age dependence in normal biomarker concentration levels was assumed to be unimportant for this investigation and a single cutoff was used for simplicity. Earliest detection time is defined to be the time at which blood biomarker concentration reaches 4 ng/mL.

For each scenario, parameter values were tuned over plausible ranges. For example, if the fraction of secreted biomarker is increased above baseline, tumor cell secretion is increased, and normal cell secretion decreased, this should favor earlier detection times. In contrast, if the fraction of biomarker reaching the blood is decreased and cancer cell secretion is decreased, this should be unfavorable for detection times. Values of parameters used in the simluations are given in Table 4. To solve the ordinary differential equations and find the timecourses of biomarker concentration in blood, numerical integration was implemented using the deSolve package in the R programming language [28].

## 5. Conclusions

In this work I found that for a PSA-like biomarker, early detection is feasible for biologically plausible parameter values. These feasible parameter regimes trend towards high tumor-to-normal biomarker secretion, gradual tumor evolution, and a high fraction of secreted biomarker entering the blood. These are intuitive conclusions, but also identify how sensitive the feasibilty of early detection is to tumor-to-normal biomarker secretion rates. This theoretical investigation urges experimental studies of biomarker secretion rates in model systems of PDAC and normal controls, where it is practical to routinely collect blood samples as disease progresses. There remains significant gaps between mathematical modeling studies such as this and wetlab experiments, which is a general problem in systems biology [32]. To help bridge the gap between dry lab and wet lab investigations, relatively simple experiments to measure biomarker secretion rates in cell lines by adapting well-established protocols used in secretome studies are recommended [33]. Preferably, measurements should be made in physical units of nanograms per milliliter (ng/mL), instead of the poorly defined units per milliliter (U/mL). Although much more costly, due to the uncertainty in biomarker penetration rates and the complex microenvironment in PDAC, I conclude that experimentation with PDAC animal models for biomarker secretion and clearance rates is warranted.

Future work in theoretical studies of biomarkers can incorporate more sophisticated models of tumor growth and biomarker secretion, including stochastic differential equation, agent-based and multiscale models, as well as models of specific cancer subtypes. These would provide mechanistic insights and a greater range of simulated behavior [20,21] to help further prioritize experimental measurements. Finally, studies such as this and Hori and colleagues [11] point to the value in measuring change in biomarkers over time; i.e., velocity; however, experience with PSA testing has shown that there is no added value for early detection in measuring PSA velocity [34]. Theoretical examination of this discrepancy is warranted. Based on the theoretical investigation presented here, the notion that the gradual evolution of PDAC will result in the feasibility of early detection with molecular biomarkers appears to be reasonable, but there is still significant uncertainty that should motivate experiments to measure biomarker secretion rates and the fraction entering blood. There are additional aspects of PDAC microenvironment, such as hypoperfusion and macropinocytosis that appear to make early detection less feasible. Finally, recent reports demonstrating excellent

sensitivity and specificity in PDAC biomarker panels provide empirical support for the feasibility of early detection, so there remain significant unanswered questions, but also optimism [35–38].

**Funding:** The author was supported by cancer Center Support Grant P30 CA008748, which provides general research support to the institution. The funding bodies played no roles in the design of the study or analysis, and interpretation of the data. This research was funded by National Institutes of Health (NIH) grant number R01 CA208401.

**Acknowledgments:** Paul Tempst and Kenneth Yu provided training, support, and encouragement.

**Conflicts of Interest:** The author declares no conflict of interest. The funder had no role in the design of the study; in the analyses, or interpretation of data; in the writing of the manuscript, or in the decision to publish the results.

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
