# Peer review of "Mathematical Modeling of The Challenge to Detect Pancreatic Adenocarcinoma Early with Biomarkers"

_challenges, doi:10.3390/challe10010026_

Round 1

Reviewer 1 Report

This is an overall interesting and well written paper.

Several points would need to be addressed:

Why was the sensitivity analysis not performed for the punctuated equilibrium model?

Can the author expand more on why CA19-9 data could not be use for modeling instead of PSA?

In Fig 1B in the formula, should the 'CancerCell' not be substituted by 'c'?

Under Methods (line 263), the formula did not come out clearly. We assume that it is the same as in Fig 1B?

Author Response

Reviewer #1

Reviewer: This is an overall interesting and well written paper. 

Response: Thank you for this kind and encouraging assessment.

Reviewer: Several points would need to be addressed: 

Why was the sensitivity analysis not performed for the punctuated equilibrium model?

Response: The equations have the same basic functional form and therefore, the general sensitivity to results to parameter variations will have the same scaling behavior.

Reviewer: Can the author expand more on why CA19-9 data could not be use for modeling instead of PSA?

Response: This is an excellent point. Despite the usefulness and routine measurement of CA19-9 in the clinic, it appears that secretion rates and clearance rates have not been published. It is hoped that by raising this issue, the article will challenge the pancreatic research community to make the measurements.

Reviewer: In Fig 1B in the formula, should the 'CancerCell' not be substituted by 'c'?

Response: Thank you for the careful and detailed reading. This error has been corrected.

Reviewer: Under Methods (line 263), the formula did not come out clearly. We assume that it is the same as in Fig 1B?

Response: Thank you for the careful and detailed reading. This error has been corrected.

Reviewer 2 Report

The author combined biomarker concentration model and tumor growth model to assess whether biomarker concentration could be an early warning signal for Pancreatic Adenocarcinoma. The mathematical model is simple but intuitive and many efforts have been made on parameter characterization and calibration. The conclusion that the ability of early detection depends on the relative secretion rates between tumor cells and normal cells make sense to me. 

I think the technical side of the paper is already at a good position. However, I am not fully convinced how many contributions this paper would add to the field of cancer research. I would suggest the author to compare the model with previous models to highlight the unique contributions of this paper. Additionally, there are three questions that may be addressed in discussions

(1) Since maths are easily transferred from system to system, can this model be used for other tumor types and what are necessary changes?

(2) Following the first question, what parameters are generic that are relatively conserved from system to system and what parameters are system-specific?

(3) Are the model parameters tumor-specific or patient-specific? If the latter, would it be possible to develop personalized model for people who want early detection?

To sum, the authors need to improve two parts of the manuscript, significance statement as well as discussions, to warrant publication.

Author Response

Reviewer #2

Reviewer: The author combined biomarker concentration model and tumor growth model to assess whether biomarker concentration could be an early warning signal for Pancreatic Adenocarcinoma. The mathematical model is simple but intuitive and many efforts have been made on parameter characterization and calibration. The conclusion that the ability of early detection depends on the relative secretion rates between tumor cells and normal cells make sense to me. 

Response: Thank you for the critical reading. When the study was begun it was unclear how the model parameterization stage would go. It gradually emerged that the PDAC community has spent a great deal of effort studying kinetics of tumor growth and evolution, and surprisingly very little work examining biomarker shedding rates. It has been said that sometimes the greatest value in making a mathematical model is not the model predictions but rather the parameters that are the least well-characterized, so that future experiments can be designed to refine models, and only then, accurate predictions can be made. 

Reviewer: I think the technical side of the paper is already at a good position. However, I am not fully convinced how many contributions this paper would add to the field of cancer research. I would suggest the author to compare the model with previous models to highlight the unique contributions of this paper. Additionally, there are three questions that may be addressed in discussions

Response: This is an excellent point and the discussion has been revised to distinguish this paper. In contrast to previous papers examining the feasibility of early detection in prostate cancer and ovarian cancer, this paper is far less optimistic.

Reviewer: (1) Since maths are easily transferred from system to system, can this model be used for other tumor types and what are necessary changes?

Response: The model is general and can be used for other tumor types. The discussion section has been improved to elaborate further (3rd paragraph of discussion)

Reviewer: (2) Following the first question, what parameters are generic that are relatively conserved from system to system and what parameters are system-specific?

Response: This is an outstanding question. Considering that there are only published models for prostate and ovarian cancer, it is unclear which parameters are relatively conserved across tumor types. Despite intense clinical interest in biomarkers, there have been relatively few studies that examined fundamental properties, such as a shedding rates. Some speculation as to differences among tumor types has been added to the discussion.

Reviewer: (3) Are the model parameters tumor-specific or patient-specific? If the latter, would it be possible to develop personalized model for people who want early detection?

Response: Thank you for this thoughtful and forward-thinking question. The model paraemters are tumor-specific. How to make them patient-specific had not even been considered during the writing of this paper and other ongoing research on biomarkers. It seems that biomarkers could be personalized in several ways. One is well-described in the literature and involves classifying individuals into high-risk groups who are screened at higher frequency. Another method might involve being able to associate genetic sequences with parameters in a biomarker model, such as perhaps certain variations in extracellular matrix genes might alter the fraction of biomarker entering blood. Or variations in biomarker gene promoter sequence might affect secretion rate by a tumor. It is conceivable that different individuals would be screened at different rates or indeed for different biomarkers depending on personalized characteristics.

Response: To sum, the authors need to improve two parts of the manuscript, significance statement as well as discussions, to warrant publication.

Response: Upon reflection of this criticism, the manuscript has been pointedly revised in the abstract and discussion to better highlight the signficance of the study and discuss the broader issues raised.

Round 2

Reviewer 2 Report

The author has addressed my questions properly and I therefore recommend its publication in present form.